# Spontaneous Mutations in *Saccharomyces cerevisiae* mtDNA Increase Cell-to-Cell Variation in mtDNA Amount

**DOI:** 10.3390/ijms242417413

**Published:** 2023-12-12

**Authors:** Elena Yu. Potapenko, Nataliia D. Kashko, Dmitry A. Knorre

**Affiliations:** 1Faculty of Bioengineering and Bioinformatics, Lomonosov Moscow State University, Moscow 119991, Russia; 2Belozersky Institute of Physico-Chemical Biology, Lomonosov Moscow State University, Moscow 119991, Russia

**Keywords:** yeast, mtDNA, genetic drift, deletions, cell-to-cell heterogeneity, cell cycle

## Abstract

In a eukaryotic cell, the ratio of mitochondrial DNA (mtDNA) to nuclear DNA (nDNA) is usually maintained within a specific range. This suggests the presence of a negative feedback loop mechanism preventing extensive mtDNA replication and depletion. However, the experimental data on this hypothetical mechanism are limited. In this study, we suggested that deletions in mtDNA, known to increase mtDNA abundance, can disrupt this mechanism, and thus, increase cell-to-cell variance in the mtDNA copy numbers. To test this, we generated *Saccharomyces cerevisiae rho^−^* strains with large deletions in the mtDNA and *rho^0^* strains depleted of mtDNA. Given that mtDNA contributes to the total DNA content of exponentially growing yeast cells, we showed that it can be quantified in individual cells by flow cytometry using the DNA-intercalating fluorescent dye SYTOX green. We found that the *rho^−^* mutations increased both the levels and cell-to-cell heterogeneity in the total DNA content of G1 and G2/M yeast cells, with no association with the cell size. Furthermore, the depletion of mtDNA in both the *rho^+^* and *rho^−^* strains significantly decreased the SYTOX green signal variance. The high cell-to-cell heterogeneity of the mtDNA amount in the *rho^−^* strains suggests that mtDNA copy number regulation relies on full-length mtDNA, whereas the *rho^−^* mtDNAs partially escape this regulation.

## 1. Introduction

Mitochondria are the semi-autonomous organelles harbouring their own genome. Mitochondrial DNA (mtDNA) is present in all eukaryotic species, with very few exceptions [1]. mtDNA encodes a few proteins which are essential for oxidative phosphorylation (OXPHOS) as well as the components of mitochondrial translation machinery: tRNAs and rRNAs [2]. mtDNA replication is considered independent from nuclear DNA (nDNA) replication, as indicated by the increase in mtDNA copy numbers upon cell cycle arrest [3,4]. At the same time, mitochondrial biogenesis and mtDNA replication are modulated by the cell cycle stages [5,6]. For instance, HeLa cells induce mtDNA replication in the S-phase and repress it during mitosis [7].

During cell division, the mitochondria and mtDNA molecules randomly segregate between newly formed cells, although this process can be constrained by the mitochondrial network structure [8]. Random segregation suggests that during the subsequent rounds of cell division, the number of mtDNA molecules in individual cells will drift. This could produce cells with depleted or abnormally elevated levels of mtDNA. Under normal conditions, however, this is rarely observed in eukaryotic cell populations, suggesting that the regulation of mtDNA amount is conducted through a negative feedback loop or active partitioning mechanisms. Such mechanisms operate with the plasmids in bacteria and yeast [9,10]. The copy number of yeast mtDNA is regulated by the balance between two nuclear-encoded HMG-box proteins, Cim1p and Abf2p, which bind mtDNA [11]. Many other genes can also regulate the mtDNA levels [12,13,14]; in particular, the antioxidant enzyme mitochondrial superoxide dismutase and the proteins of the mitochondria-to-nucleus signalling Rtg cascade negatively contribute to the maintenance of mtDNA under stress conditions [15]. Moreover, mtDNA copy number regulation in yeast is mediated by the oxidative stress levels [16]. Despite this knowledge, the mechanisms regulating the mtDNA copy number, and particularly those preventing mtDNA copy number drift, largely remain unknown [17].

Eukaryotic cells usually contain multiple copies of mtDNA molecules, while the mitochondrial DNA repair mechanisms are limited [18]. These two factors make mtDNA more prone to mutations and deletions. Indeed, the human germline mtDNA mutation rate is an order of magnitude higher than the mutation rate of nDNA [19], and mtDNA with large deletions are accumulated as humans age [20]. At the same time, mutant mitochondrial DNAs with deletions or other deleterious mutations can accumulate at a high level (copies per cell) and displace parental (wild-type) mtDNAs [21]. Many yeast strains are prone to mutations in mtDNA, and their cultures usually have a high proportion (up to several percent) of cells with the *petite* phenotype [22]. *Petite* cells can neither respire nor utilise non-fermentable carbon sources. Importantly, the *petite* phenotype is associated with the mutations in mitochondrial DNA, most of which are large-scale deletions [23,24,25]. Yeast mutant strains with large deletions in the mtDNA accumulate mtDNA molecules in cells in large amounts [26].

In this study, we hypothesised that mtDNAs with large deletions escape copy number control mechanisms, and therefore, the populations of cells with mutant mtDNA are more heterogeneous in the mtDNA content than the populations of cells with wild-type mtDNA. To test this hypothesis, we studied several *S. cerevisiae* strains with spontaneous large deletions in mtDNA and quantified the level of variation in the total DNA amount in them. *S. cerevisiae* cells have a compact nuclear genome of ~12 Mb and a relatively large mitochondrial genome of approximately 85 kb [27]. As a result, in yeast, mtDNA represents a substantial fraction of the total cellular DNA, from 15 to 25% [28]. Therefore, mtDNA can be detected with flow cytometry using standard DNA-intercalating fluorescent dyes. Moreover, in the *S. cerevisiae* laboratory strains, the rate of spontaneous mtDNA deletions is high, and therefore, the cell suspensions usually contain a fraction of the cells that retain only a partial mitochondrial genome [22,29]. Mutant strains with large deletions and the corresponding mitochondrial genotypes are referred to as *rho^−^* and can be easily obtained and distinguished from parental *rho^+^* strains containing the full-length mitochondrial genome.

Here, we analysed the cell-to-cell heterogeneity in the total DNA levels in suspensions of a wild-type (*rho^+^)* yeast strain and strains with large deletions in mitochondrial genomes (*rho^−^*). We used the *rho^0^* strain lacking mtDNA as a control. We also analysed a previously obtained hypersuppressive (*HS) rho^−^* strain that retained only a short 2 kb mitochondrial genome [30]. Furthermore, we examined the variation in the total DNA levels in the ‘nuclear petite’ strains, which were lacking one of the genes essential for oxidative phosphorylation. The variation was analysed separately for the G1 and G2/M cells and for varying size classes of cells.

## 2. Results

To test whether the mtDNAs with large deletions escape copy number regulation, from a haploid *rho^+^* strain we generated spontaneous *rho^−^* strains and a *rho^0^* strain completely lacking mitochondrial DNA (Appendix A). We cultivated these strains in batch cultures up to the exponential phase and stained the cell suspensions with the DNA-intercalating dye SYTOX green (see Section 4). All the strains showed a bimodal distribution of SYTOX green intensities that correspond to the cells with unduplicated and duplicated genomic DNA, 1n and 2n (Figure 1A). In the *rho^0^* cell suspensions, the positions of the modes of both peaks were shifted to the left (Figure 1A,B). We obtained consistent results using the *cdc4-3 rho^+^* and *rho^0^* strains; these strains have a cell cycle defect that manifests at elevated temperatures. Under the conditions of cell cycle arrest, both strains exhibited an increase in the proportion of 1n cells. At the same time, the distributions of SYTOX green signal intensities were shifted to higher values in *cdc4-3 rho^+^*, but not in *cdc4-3 rho^0^* yeast cells (Appendix A). This confirmed that mtDNA contributes to the integral SYTOX green signal and can be detected using flow cytometry.

Remarkably, the cells from all six independently generated *rho^−^* strains exhibited higher SYTOX green signals in both cell cycle stages. Similar results were obtained for the *HS rho^−^* strain (Figure 1B). At the same time, the *rho^−^* cells have a negligible difference in the growth rates under normal conditions compared to those of the *rho^0^* cells, whereas the *rho^+^* cells exhibited higher proliferation rates (Appendix A). Given that the cell cycle duration depends mainly on the G1 phase duration, while the lengths of other phases are more uniform, the proportion of G1 (1n) to G2/M (2n) cells in the *rho^−^* and *rho^0^* strains was higher than that in the parental *rho^+^* strain, which showed a higher growth rate (Figure 2).

Next, we analysed the variation in SYTOX green fluorescence intensities for the *rho^+^*, *rho^−^*, and *rho^0^* cells. To evaluate the SYTOX green signal representing mtDNA, we needed to analyse cells with different amounts of nDNA separately. Therefore, to ensure that the contributions of yeast cells in different stages of the cell cycle did not distort the results, we analysed only the right side of the distribution peak for the G2/M cells and the left side of the peak for the G1 cells (Appendix A). This reduced the number of analysed cells in each sample, but assured that all cells were in the same cell cycle phase and contained the same amount of nDNA. We then calculated the robust coefficients of variation (rCV, see Appendix A for the equations) for the SYTOX green signal in the 1n and 2n subpopulations. To mitigate any potential day-to-day variations in the replicas, we normalised each calculated rCV value against the control *rho^+^* strain’s rCV value from the same batch. Figure 3 demonstrates that 1n and 2n cell subpopulations of *rho^+^* strains exhibit lower rCV values than the corresponding 1n and 2n subpopulations of *rho^−^* strains. Concurrently, the cells of the *rho^0^* strains were more uniform in DNA content; the *rho^0^* strains showed lower rCV values than the population of parental *rho^+^* cells (Figure 3). Furthermore, increased cell-to-cell heterogeneity of the SYTOX green signals in the *rho^−^* cell populations compared to those of the *rho^+^* cell populations was also observed when the yeast cells were cultivated at 25 °C instead of 30 °C (Appendix A). Under these conditions, the cells exhibit an increase in the duration of the G1 cell cycle stage and a larger proportion of G1 cells.

We suggested that the increase in SYTOX green signal variation could be associated with the increased cell size variation of *rho^−^* cells. Therefore, we analysed how the SYTOX green mean intensity and cell-to-cell variation depend on the cell size. In order to achieve this, we took the events (cells) with SYTOX green values above the mode of the G2 (2n) peak (see Appendix A for automatic identification of a mode in 2n peak position). Then, we plotted the distributions of the forward scattering areas (FSC-A) of such cells and identified the positions of distribution modes for each experiment (Figure 4A and Appendix A). After this, we calculated the SYTOX green signal intensities rCV for varying window sizes, centred on the FSC-A distribution mode (Figure 4A, upper panel). We found that narrowing the window size reduces the rCV to a certain threshold (Figure 4B). This threshold is commonly referred to as intrinsic variation (noise), distinguishing it from extrinsic variation, which depends on the cell size [31]. A narrow FSC-A window size represents the yeast cells with similar sizes; nonetheless, the *rho^−^* cells exhibited increased SYTOX green rCV values, even if compared within narrow cell size windows (Figure 4B). In addition, we found that the yeast mitotypes exert a minimal, if any, effect on the cell-to-cell variation in mitochondrial protein abundance. This was demonstrated using yeast strains that express Idh1-GFP, a mitochondrial isocitrate dehydrogenase fused with GFP. The variance in Idh1-GFP levels was consistent across *rho^−^*, *rho^0^*, and *rho^+^* strains (Appendix A).

To test if the increase in SYTOX green rCV values is associated with the inability of yeast cells to perform oxidative phosphorylation, we examined the variation in the SYTOX green signal in the ‘nuclear petites’. These are yeast strains that lack one of the nuclear-encoded OXPHOS genes and, as a result, are incapable of utilising non-fermentable carbon sources. We took three ‘nuclear petite’ strains of the *W303* genetic background: a strain with a deleted *COQ3* gene that encodes an enzyme required for coenzyme Q_6_ biosynthesis, and two *cyt1Δ* strains with different prototrophic markers. *CYT1* is a catalytic subunit of respiratory complex III. We confirmed that all of these mutants are unable to utilise glycerol, but retain mtDNA (see Section 4). However, unlike the *rho^−^* cells, we did not detect an increase in the SYTOX green signal rCV of these strains, compared to that of the parental *rho^+^* strain (Appendix A).

Next, we evaluated how the SYTOX green signal depends on the cell size. To achieve this, we calculated the mean and rCV of SYTOX green signal intensities for the yeast cell populations clustered by cell size. We found that the mean SYTOX green intensities increase with cell size in the *rho^+^* and *rho^−^* strains to a much greater extent than those in the *rho^0^* strains (Figure 4A, middle panel). The slopes of linear regression between the SYTOX green signal and FSC-A bin size were 0.94 ± 0.28 for *rho^+^*, 0.80 ± 0.40 for *rho^−^*, and 0.20 ± 0.08 for the *rho^0^* cell suspensions (*p* < 0.001 if compared to the *rho^+^* and *rho^−^* slopes according to the Wilcoxon rank-sum test). The subpopulations of large *rho^−^* and *rho^+^* cells also showed higher SYTOX green rCV values (Figure 4A, lower panel), while the SYTOX green rCV values of *rho^0^* cells remained almost constant. Finally, we analysed the variation in cell volume of the *rho^+^*, *rho^−^*, and *rho^0^* strains. In order to achieve this, we approximated the cell volume as FSC-A multiplied by forward scattering intensity height (FSC-H) and calculated the rCV for each experiment. In contrast to the SYTOX green signal, we did not find any relationship between the mitochondrial DNA genotype and cell volume (Figure 4C).

## 3. Discussion

In this study, we have demonstrated that, in yeast, the amount of mtDNA can be assessed at the level of individual cells in the G1 and G2/M cell cycle stages using a standard flow cytometry technique. As depicted in Figure 1, the *rho^+^* and *rho^−^* cells exhibit a higher integral SYTOX green signal intensity compared to that of the *rho^0^* cells. This observation indicates that, in yeast cells, mtDNA contributes significantly to the total SYTOX green signal. However, it is important to note that yeast cells may contain extrachromosomal elements other than mtDNA, and the total DNA content could be substantially influenced by changes in the individual chromosome copy numbers (aneuploidies). Approximately 20% of laboratory, industrial, and natural yeast strains contain aneuploidies [32]. Nonetheless, the duplication and loss of chromosomes are rare; the rate of chromosome loss is about 10^−7^–10^−6^ events per cell division [33,34], and thus, it is unlikely to contribute to the cell-to-cell variation in the total DNA content of the yeast suspensions evaluated in this study. On the other hand, yeast cells can contain 40–60 molecules of multicopy 2µ plasmids, each with a size of 6.3 kb [35]. These could potentially contribute to the overall yeast DNA content. However, even at its maximum, the total plasmid DNA in yeast is equivalent to the size of approximately 4–5 mtDNA molecules. Furthermore, the copy number of 2µ plasmids is regulated by negative feedback loop mechanisms encoded within the plasmid itself [35]. Therefore, multicopy plasmids should not make a significant contribution to the total DNA amount and variation. Lastly, individual cells could harbour extrachromosomal rDNA circles (ERCs). ERCs are excised from rDNA tandem repeats composed of multimer rDNA repeat units of 9.6 kb and are accumulated in old yeast cells [36]. Theoretically, ERCs can significantly contribute to the total DNA content of individual cells. However, in an exponentially growing yeast culture, yeast cells are continuously diluted by the new daughter cells, and thus, old mother cells represent a very small portion of the total cell population [37]. Therefore, we can assume that when considering cells in identical cell cycle stages, the variation in the total DNA amount is predominantly influenced by the variation in the number of mtDNA molecules per cell.

Our study aimed to test the hypothesis that asynchronous liquid cultures of yeast cells with mutant *rho^−^* mtDNA are more heterogeneous in the cellular DNA content than the cultures of cells with *rho^+^* wild-type mtDNA. The results, as shown in Figure 3, confirm this hypothesis: in almost all the experiments, the SYTOX green signal intensity rCV values of the *rho^−^* strains were above the corresponding rCV values of the parental *rho^+^* strain. Furthermore, when we examined the SYTOX green rCV values for different cell sizes, we found that this result remains consistent (Figure 4). Thus, an increase in the SYTOX green rCV in *rho^−^* cell suspensions cannot be attributed to the variation in cell sizes between the *rho^−^* and *rho^+^* strains. On the contrary, the increase is a consequence of the mutant mitotype, regardless of the cell size.

Next, we compared how the SYTOX green fluorescence changes across varying cell sizes. In *rho^0^* cell suspensions, the mean SYTOX green signal values remained constant in the sliding FSC-A windows (Figure 4A, middle and lower panels). This suggests that the cell parameters, such as cell size and number of vacuoles, do not significantly contribute to the SYTOX green signal. At the same time, we found that in both the *rho^−^* and *rho^+^* strains, the larger cells exhibit a higher mean SYTOX green signal and greater variation compared to those of the smaller ones (Figure 4A, middle and lower panels). These observations imply that mtDNA contributes to the total DNA content in the cells in the same cell cycle phase. The increase in SYTOX green signal values suggests that as the cell volume increases, so does the mtDNA amount. The scaling of mtDNA abundance with yeast cell volume is consistent with the previous findings employing quantitative PCR [38]. Meanwhile, the increase in SYTOX green signal variation in the large cells is potentially attributable to mitochondrial dysfunction in the replicatively old cells [39], which are more prevalent in large-sized bins, whereas small-sized bins are supposedly composed mainly of newborn daughter and young mother cells.

Why do *rho^−^* strains show a high cell-to-cell variation in mtDNA content? Having a large deletion in mtDNA makes these cells incapable of oxidative phosphorylation. This, in turn, can disrupt the hypothetical negative feedback loop that regulates the mtDNA copy number. In this study, we also tested yeast cells that cannot respire: a strain with blocked coenzyme Q_6_ biosynthesis, an essential cofactor of the respiratory chain [40], and without the core component of respiratory complex III, Cyt1p [41]. These ‘nuclear petite’ strains exhibited neither an increased SYTOX green level nor an increase in the SYTOX green intensity population variance (Appendix A). This suggests that the regulation of the mtDNA copy number is not directly related to the functioning of the respiratory chain. At the same time, we cannot completely rule out the possibility that proteins encoded in the mtDNA regulate the mtDNA copy number. For example, their regulatory role may be realised not by their enzymatic function, but by the abundance of certain mtDNA-encoded proteins or the general activity of mitochondrial transcription or translation processes, which are inhibited in the *rho^−^* cells.

Another possibility is that the original deletion in the mtDNA increases the probability of secondary deletions, producing a variety of mitochondrial genomes present in the cell suspension [29]. This variety can be manifested in the increased total DNA amount heterogeneity if the mitogenomes significantly vary in copy number and size.

Finally, an increase in mtDNA content variance could be due to the concatenation of mutant DNA molecules associated with a decrease in the copy number of the concatenates in the *rho^−^* cells. Indeed, despite *rho^−^* mtDNAs being shorter, they can be concatenated into larger mtDNA molecules [42]. Consequently, they represent a smaller number of mtDNA segregation units than *rho*^+^ mtDNA. A decrease in the number of mtDNA segregation units could intensify the drift in copy numbers, increasing the population variance in the mtDNA amount.

Regardless of which mechanism or combination of mechanisms contributes to the high cell-to-cell variation in mtDNA amount, our results show that the deletions in mtDNA increase the cell-to-cell heterogeneity in the mtDNA amount. Additionally, our findings suggest that for yeast cells with mutated mtDNA, it is difficult to maintain a normal mtDNA/nDNA ratio. While it is important to note the significant differences in mtDNA genome organisation and maintenance machinery between fungi and metazoans, we propose that the deterioration of copy number regulation control in cells with mutant mtDNA could be a widespread mechanism contributing to mitochondrial pathologies in various species, including humans.

## 4. Materials and Methods

### 4.1. Yeast Strains and Growth Conditions

In this study, we used the mat *alpha W303-1B* laboratory strain (*MATalpha ade2-101 his3-11 trp1-1 can1-100 leu2-3*) as well as *rho^−^* and *rho^0^* derivative mutants. The relationship between parental and derived strains is indicated in Appendix A. We also utilised the *cdc4-3* thermo-sensitive mutant strain from the ts-collection [43], along with its *rho^0^* derivative mutant. For the analysis of mitochondrial protein abundance, we used a set of *W303 IDH1-GFP rho^−^*, *rho^+^*, and *rho^0^* strains. These strains had previously been generated in our laboratory [30,44].

To estimate the growth rates, we inoculated yeast strains in YPD (1% yeast extract; 2% bactopeptone; 2% D-glucose) in 96-well plates (200 µL in a well, starting density 10^6^ cells/mL that corresponds to OD_550_ ~0.02) and incubated them in a SpectroStar Nano (BMG Labtech GmbH, Ortenberg, Germany) microplate spectrophotometer at 30 °C. The optical density (OD_550_) was assessed every 5 min for the entire duration of the experiment.

We also utilised the mat *a W303-1A cyt1Δ::HIS3*, *cyt1Δ::TRP1*, *and coq3Δ::HIS3* strains, which are incapable of growing on non-fermentable carbon sources. To confirm the retention of mtDNA in these strains, we crossed them with the *MATalpha rho^0^* strain and subsequently plated the crossing on non-fermentable carbon source plates, YPGly: 2% bactopeptone, 1% yeast extract, and 2% glycerol. While both parental strains could not utilise glycerol as the sole carbon source, the generated diploid strain did (Appendix A). This control experiment demonstrated that, in spite of the absence of nuclear-coded OXPHOS genes, all three tested strains retained mtDNA.

### 4.2. mtDNA Depletion

To generate *rho^−^* strains, we cultivated the parental *rho^+^* strain on a YPDGly plate containing 1% yeast extract; 2% peptone; 2% glycerol; and 0.1% D-glucose and isolated small colonies. We ensured that strains originating from small colonies were unable to grow on a non-fermentable carbon source, but retained a signal of DNA intercalating agent in the cytoplasm, similar to the parental *rho^+^* cells. To generate *rho^0^* strains, we incubated the parental *rho^+^* or *rho^−^* strains overnight at 30 °C in YPD supplemented with ethidium bromide (0.1 mg/mL). The cell suspension was then plated on solid YPD medium. To discriminate between the *rho^−^* and *rho^0^* mitochondrial genotypes, the cells were fixed in 70% (*v*/*v*) ethanol and stained with 2 μg/mL DAPI (4′,6-diamidino-2-phenylindole, Lumiprobe, Hunt Valley, MD, USA) [30]. We checked that the resulting cells showed no DAPI signal outside of the nucleus (Appendix A).

### 4.3. Flow Cytometry

Yeast cells were grown at 30 °C under rotary shaking (250 rpm) to a density of 2 × 10^6^ cells/mL in liquid YPD medium prepared as described by Sherman [45]. After growth, the cells were harvested by centrifugation, and then resuspended in 1 mL of 70% (*v*/*v*) EtOH. The fixed cells were stored at −20 °C.

The protocol for total DNA content analysis using SYTOX green (SYTOX^TM^ green, ThermoFisher Scientific, Waltham, MA, USA) is a modification of the cell cycle analysis protocol described by Haase et al. [46]. We transferred ~1.5 × 10^6^ ethanol-fixed cells to a microfuge tube and washed them twice in 400 µL sodium citrate buffer (50 mM, pH = 7.0). RNAse was added to the final concentration, 0.3 mg/mL, and the mix was incubated for 2 h at 37 °C. It is important to eliminate RNA because DNA stains can also bind to RNA. After this, proteinase K (0.2 mg/mL, Amresco 0706, Solon, OH, USA) was added, and cells were incubated for one hour at 50 °C. This step helps to improve the access of stain to DNA. Cells fixed in ethanol tend to agglomerate, so we sonicated the samples before staining to separate the cell clumps. SYTOX green was added to the final concentration of 0.625 µM.

Cellular fluorescence from SYTOX green was determined quantitatively using a CytoFLEX flow cytometer (Beckman Coulter, Brea, CA, USA). The laser wavelength was 488 nm, and the emission filter was 525/40 nm bindpass (FITC-A). For each sample, 5 × 10^4^ cells were analysed, and the flow rate was set to 10 µL/min.

### 4.4. Data Analysis

The flow cytometry data were analysed with CytExpert software, version 2.0 (Beckman Coulter) and R environment using tidyverse [47]. We imported the flow cytometry experiment data using the flowCore [48] bioconductor package without using log transformation. Then, we identified the position of the 2n peak mode in SYTOX green signal intensity distribution using the multimode [49] package. In order to improve automatic peak mode identification, we trimmed the FSC-A and SYTOX green signal data by withdrawing 1% extreme data points. Thereafter, we calculated the mean SYTOX green intensities and the rCV for the varying FSC-A bins.

Whenever possible, in the figures, we show separate data points which represent the flow cytometry experiments from separate batch cultures.

## Figures and Tables

**Figure 1 ijms-24-17413-f001:**
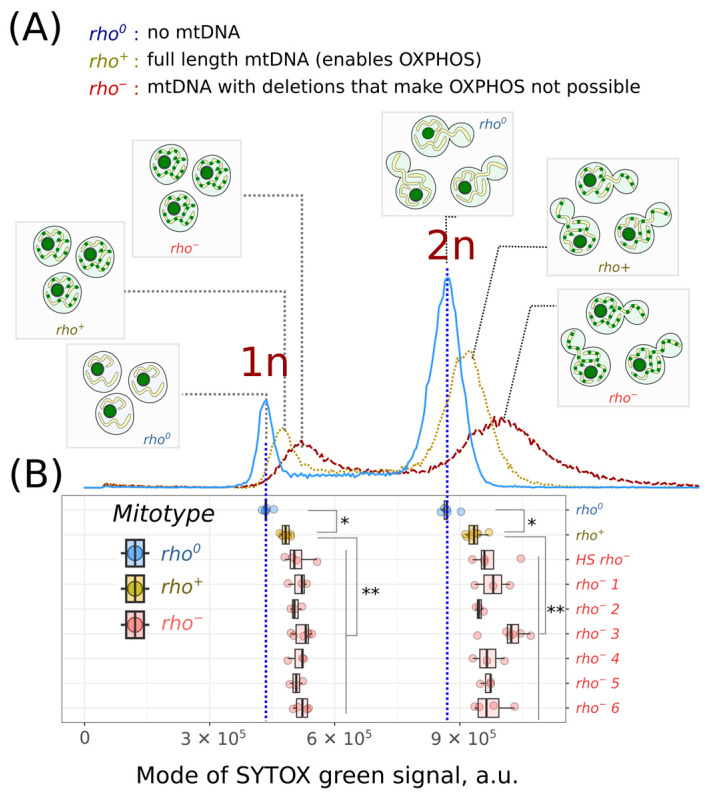
Mitochondrial DNA contributes to the SYTOX green signal in yeast. (**A**) Representative flow cytometry diagram representing the SYTOX green intensity signal of *rho^0^*, *rho^+^*, and *rho^−^ S. cerevisiae* cells. Insets depict yeast cells in G1 and G2/M phases with different mitotypes, highlighting both the nucleus and mitochondria. These components contribute to the SYTOX green fluorescence. (**B**) Local maxima of the SYTOX green signal distributions in individual experiments. * *p* < 0.05, and ** *p* < 0.005, according to the Wilcoxon rank-sum exact test with Bonferroni adjustments.

**Figure 2 ijms-24-17413-f002:**
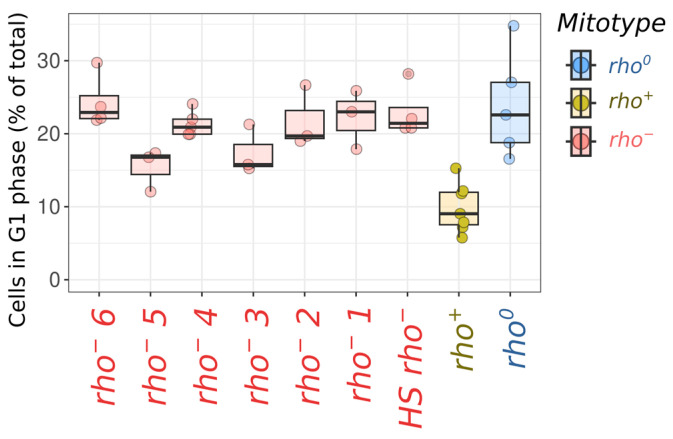
Proportion of G1 cells in *rho^−^*, *rho^+^*, and *rho^0^* cell suspensions.

**Figure 3 ijms-24-17413-f003:**
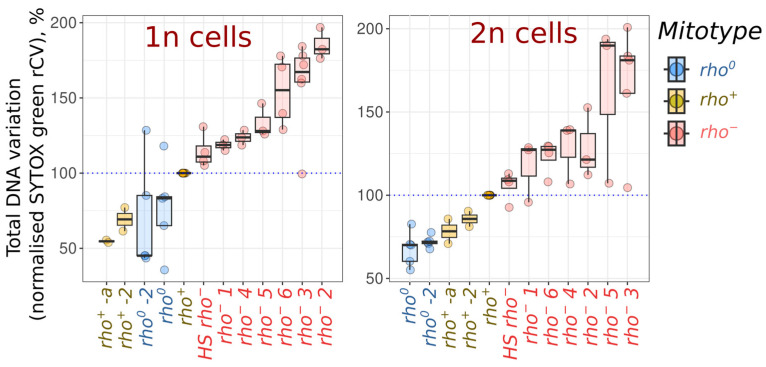
*rho^−^* strains show increased variation in total DNA content of 1n (G1 phase) and 2n (G2/M) yeast cells. For G1 (1n) and G2/M (2n) cells, *p*-values were less than 0.02 for comparisons of rCV values of both *rho^0^* and *rho^−^* strains with parental control (Wilcoxon rank-sum exact test with Bonferroni adjustments). Strains are ordered according to the mean normalised rCV values. *rho^+^*-2, *rho^+^*-a, and *rho^0^*-2 are independently obtained *rho^+^* and *rho^0^* strains (see Appendix A for details).

**Figure 4 ijms-24-17413-f004:**
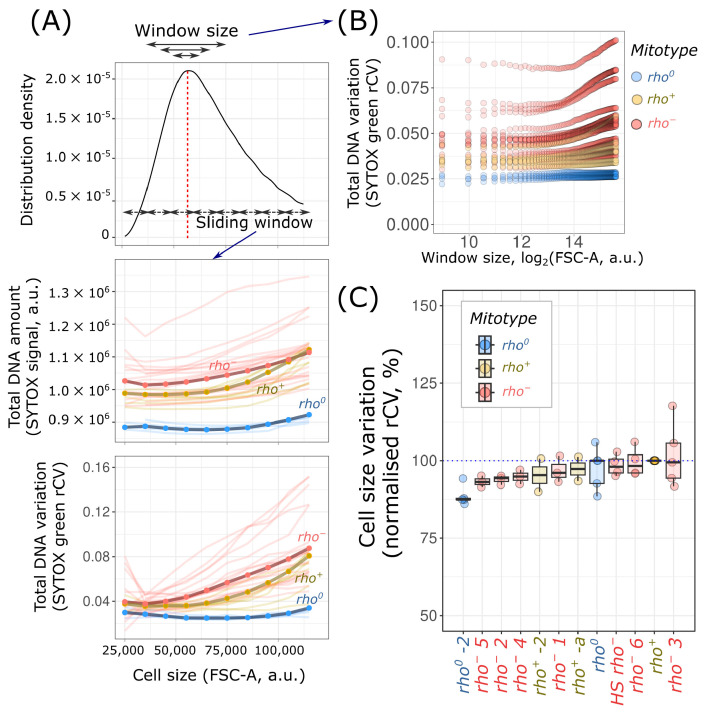
High variation of SYTOX green signal intensity in *rho^−^* cell populations is consistent across different cell sizes. (**A**) Upper panel: yeast cell forward scattering amplitude (FSC-A) distribution; middle panel: average SYTOX green intensities as a function of cell size; lower panel: rCV of SYTOX green intensities as a function of cell size. FSC-A bin size was set to 10,000 a.u. Bold lines represent average values; dim lines represent individual experiments; (**B**) rCV across different cell size ranges. The *rho^+^* and *rho^−^* rCV values, calculated specifically for narrower size bins with log_2_(FSC-A) below 12, show a significant difference, with a *p*-value of 0.0078 according to the Wilcoxon rank–sum exact test; (**C**) Cell size variation in *rho^+^*, *rho^0^*, and *rho^−^* strains.

## Data Availability

All data are present within the manuscript and Appendix A.

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
