# Peer review of "Spontaneous Mutations in Saccharomyces cerevisiae mtDNA Increase Cell-to-Cell Variation in mtDNA Amount"

_ijms, 2023, doi:10.3390/ijms242417413_

Round 1
Reviewer 1 Report
Comments and Suggestions for Authors
The manuscript by Potapenko et al describes the variation of mtDNA distribution in yeast strains with large deletion of mtDNA. The authors exploit cytofluorimetry and different yeast strains (rho+, rho- and rho0) to analyze a correlation between variation in mitochondrial DNA (mtDNA) and total DNA levels in the yeast cells (mtDNA / nDNA).
This manuscript presents many flaws as detailed below.
1) The authors have already published a paper in which some of the same yeast strains used in this work have already been analyzed. In the published paper (Mitochondrial depolarization in yeast zygotes inhibits clonal expansion of selfish mtDNA. https://doi.org/10.1242/jcs.197269) they also investigate mtDNA by techniques that led to results that solidly explain what the authors claim.
On the contrary, the results achieved in this manuscript are not fully justified experimentally.
In particular, the data obtained with the rho- strains, where the authors show higher cell-to-cell variation in mtDNA content than in rho+ strains, are only justified with a set of assumptions, leaving many questions unanswered (Discussion section: lines 213 to 234).
2) Several points are cryptic and unclear, and this hinders the fully understanding of the manuscript message to those who are not expert in the yeast field (and this journal has a general reading). For example, which deletion the strain rho- carries on? Do the authors verify the deletion by sequencing? If yes, how? What is the precise relationship between the cell cycle phases and the mtDNA replication/amount? In my opinion, authors should briefly answer these questions in the Introduction.
3) The main technical flaw is that the measurement of mtDNA amount cannot be considered reliable for at least two reasons. First, the mitochondrial mass was not calculated and this represents my main concern. It is not clear how the authors may exclude that any variation in mtDNA is not due to variation in mitochondrial mass. Nonetheless, mitochondrial mass can be measured by flow cytometry, with the use of specific fluorescent probes, and by Western blot, by analyzing several mitochondria markers. Second, the plasmid 2m might potentially contribute in the total DNA quantification. To exclude this hypothesis, authors should verify that the amount of plasmid DNA is not affected by the various treatments.
Overall, the manuscript appears as a mere technical article where authors explain how to apply flow cytometry to the analysis of mtDNA distribution/ratio. This, however, is not a flaw per se, but it is not appropriate for a Research article.
Comments on the Quality of English Language
The quality of english language is acceptable.
Reviewer 2 Report
Comments and Suggestions for Authors
Despite 50+ years of research on mtDNA in yeast, the factors that control mtDNA copy number are still not well understood. This manuscript describes a flow cytometry experiment showing that rho- strains have a higher total DNA (ie mtDNA) content than rho+ strains, that there is variation in the total mtDNA content between different rho- strains, and that there is greater cell-to-cell variation (heterogeneity) of mtDNA content within rho- strains. The authors suggest that the rho- strains have interrupted a feedback loop that regulates mtDNA content. These findings are exciting and offer insight into mtDNA regulation, and the manuscript is well written and a pleasure to read. Making connections to other work on mtDNA regulation and copy number would increase the significance of the work.
Major comments
1. There is variation in the total DNA content (Fig 1B) and rCV (Fig 3/S4) between the different rho- strains, and this suggests that it might be possible to map mtDNA sequences that interrupt the proposed feedback loop. This could be done by characterizing the mtDNA sequences in the rho- strains used here, or by performing this experiment on previously characterized rho- mtDNAs. This would greatly increase the importance of the current work and may offer a way to connect these results with other studies on rho- mtDNAs (such as how different lengths of rho- mtDNAs influence nucleoid spacing- which may be related to the instability observed here).
2. A limitation of this work is that it assumes that differences in total DNA fluorescence between rho+/-/0 strains are due to differences in mtDNA content, and therefore are indirect measures. Why not use Sybr Green or another mtDNA-specific dye for these experiments? If additional validation is not performed, the authors should include a discussion explaining why more direct measures of mtDNA content were not performed.
3. This manuscript is written for readers who understand yeast mitochondrial genetics. General readability could be improved. (add explanations of rho+/- (and probably avoid this terminology in the title), HS petites, etc.)
4. Many figures are in red and green; individuals that are red/green colorblind may have difficulties in reading them. Please verify that the color scheme is accessible.
Minor points:
5. Fig1: While the trace for the rho0 strain in 1A lines up with the median line in the rho0 boxplot in Fig 1B, the trace for a rho- strain doesn’t appear to match any of the data shown in Fig 1B. Maybe I am misinterpreting this? All comparisons are made against the distributions in a rho+ strain. It would be helpful to show a trace of the rho+ strain here.
6. Referring to Figure 3, the authors state that the rho0 strains were “more uniform in DNA content”, (line 110). The 1n populations for the rho0 strains show some of the largest distributions of normalized rCV (meaning size of the boxes in the box and whisker plots). Can the authors clarify what they mean by this?
7. Fig S4: title says rho- strains show increased rCV of total DNA. It doesn’t look like all the rho- strains have increased rCV% relative to rho+ strain. Maybe this can be addressed by stating “Some rho- strains” in the title, and also in the main text (line 112- “in some rho- populations).
8. Line 114: Do you mean to refer to FigS4 for rCV comparisons and Fig S5 for a comparison of cells in G1 phase?
9. The authors include thoughtful discussion of other factors that may contribute to fluorescence differences (aneuploidies, plasmids, etc.). Vacuoles may absorb fluorescence, and this could contribute to differences in fluorescence, particularly in large, older cells. This could be added to the discussion.
10. Lines 301-311 should be removed.
Reviewer 3 Report
Comments and Suggestions for Authors
The manuscript “Cell-to-cell variation of mtDNA amount is higher in rho- than in rho+ yeast strains” by Potapenko et al. deals with the homeostasis between the nuclear and the mitochondrial genomes in eukaryotic cells and its regulatory mechanisms. The authors use S.cerevisiae yeast cells as model and suggest that deletions in mtDNA, known to increase mtDNA abundance, can disrupt this mechanism and, thus, increase cell-to-cell variance in mtDNA copy numbers. The topic is highly interesting and suffers of sufficient experimental evidences. This work is a valuable attempt to address certain points on the topics, however I have several issues/comments which need to be addressed before publication.
-Figure 1.
Why the diagram referred to rho+ cells is not shown? It would help to understand the results.
About the 2n distribution, are the authors able to distinguish between budding cells or agglomerate of single cells?
Fig.1A. Rho0 and rho- cells show circular green molecules and linear molecules without colour or with interspersed green colour. Can the authors clarify this point? This will help to differentiate rho0 and rho- cells.
-To evaluate the SYTOX signal representing mtDNA, the authors analyzed cells with different amounts of nDNA separately. Why did not they use synchronyzed cultures?
-Lack of oxidative phosphorylation in rho- cells should be associated to increased oxidative stress. Did the authors have measured intracellular ROS level in these cells? A comment on this aspect should be included related to the mechanism of regulation on mtDNA copy number.
Figs. 3, 4, S4, the value for rho+ is set as control, did the authors observe differences in the absolute values of the control between G1 and G2/M phases?
The list of references might be improved with more recent literature on the topic.
Minor points:
-Line 13, “known to increase mtDNA abundance” should be closed by two commas
-Please, check the punctuation along the whole manuscript
-Fig.S1 Can the authors clarify the meaning of HS rho-?
-Fig.S2 Growth conditions should be indicated in the legend.
-Fig.S3, make a correction for subpopulation
-Fig.S5 Is it possible to have an overlay of rho+, rho- and rho0 as a representative experiment?
-Fig.S7 When the cells were collected for this analysis?
A) please, indicate the peaks (G1 or G2/M)
B) the legend referred to the blu symbol (rho0) is missing
-Lines 203-206 of the Discussion, the two sentences seem contradictory. Please clarify.
-The indication of “rho- and rho+” strains in the title might limit the number of readers for this article. It would be desirable to choose a more general title.
Comments on the Quality of English LanguageMinor editing of English language required. Punctuation has to be checked throughout the manuscript.
